# Acute Heart Failure: Diagnostic–Therapeutic Pathways and Preventive Strategies—A Real-World Clinician’s Guide

**DOI:** 10.3390/jcm12030846

**Published:** 2023-01-20

**Authors:** Ciro Mauro, Salvatore Chianese, Rosangela Cocchia, Michele Arcopinto, Stefania Auciello, Valentina Capone, Mariano Carafa, Andreina Carbone, Giuseppe Caruso, Rossana Castaldo, Rodolfo Citro, Giulia Crisci, Antonello D’Andrea, Roberta D’Assante, Maria D’Avino, Francesco Ferrara, Antonio Frangiosa, Domenico Galzerano, Vincenzo Maffei, Alberto Maria Marra, Rahul M. Mehta, Rajendra H. Mehta, Fiorella Paladino, Brigida Ranieri, Monica Franzese, Giuseppe Limongelli, Salvatore Rega, Luigia Romano, Andrea Salzano, Chiara Sepe, Olga Vriz, Raffaele Izzo, Filippo Cademartiri, Antonio Cittadini, Eduardo Bossone

**Affiliations:** 1Cardiology Division, A. Cardarelli Hospital, Via Cardarelli, 9, 80131 Naples, Italy; 2Department of Advanced Biomedical Sciences, University of Naples Federico II, Via Sergio Pansini, 5, 80131 Naples, Italy; 3Department of Translational Medical Sciences, Federico II University, 80131 Naples, Italy; 4First Aid—Short Intensive Observation Division, A. Cardarelli Hospital, Via Cardarelli, 9, 80131 Naples, Italy; 5Emergency Medicine Division, A. Cardarelli Hospital, Via Cardarelli, 9, 80131 Naples, Italy; 6Unit of Cardiology, Department of Translational Medical Sciences, University of Campania “Luigi Vanvitelli”, Monaldi Hospital, 80131 Naples, Italy; 7Long-Term Care Division, Cardarelli Hospital, Via Cardarelli, 9, 80131 Naples, Italy; 8Istituto di Ricovero e Cura a Carattere Scientifico SYNLAB SDN, Via Emanuele Gianturco, 113, 80143 Naples, Italy; 9Heart Department, University Hospital of Salerno, 84131 Salerno, Italy; 10Department of Cardiology, Umberto I Hospital Nocera Inferiore, 84014 Nocera, Italy; 11Post Operative Intensive Care Division, A. Cardarelli Hospital, 80131 Naples, Italy; 12Heart Centre, King Faisal Specialist Hospital and Research Centre, Riyadh 11211, Saudi Arabia; 13ProMedica Monroe Regional Hospital, Monroe, MI 48162, USA; 14Duke Clinical Research Institute, 300 W Morgan St., Durham, NC 27701, USA; 15Department of Public Health University “Federico II” of Naples, Via Sergio Pansini, 5, 80131 Naples, Italy; 16Department of General and Emergency Radiology, Antonio Cardarelli Hospital, Via Cardarelli, 9, 80131 Naples, Italy; 17Technical Nursing and Rehabilitation Service (SITR) Department, Cardarelli Hospital, 80131 Naples, Italy; 18Department of Radiology, Fondazione G. Monasterio CNR-Regione Toscana, Via Moruzzi, 1, 56124 Pisa, Italy

**Keywords:** acute heart failure, biomarkers, cardiac ultrasound, computer tomography, therapeutic interventions, preventive strategies

## Abstract

Acute heart failure (AHF) is the most frequent cause of unplanned hospital admission in patients of >65 years of age and it is associated with significantly increased morbidity, mortality, and healthcare costs. Different AHF classification criteria have been proposed, mainly reflecting the clinical heterogeneity of the syndrome. Regardless of the underlying mechanism, peripheral and/or pulmonary congestion is present in the vast majority of cases. Furthermore, a marked reduction in cardiac output with peripheral hypoperfusion may occur in most severe cases. Diagnosis is made on the basis of signs and symptoms, laboratory, and non-invasive tests. After exclusion of reversible causes, AHF therapeutic interventions mainly consist of intravenous (IV) diuretics and/or vasodilators, tailored according to the initial hemodynamic status with the addition of inotropes/vasopressors and mechanical circulatory support if needed. The aim of this review is to discuss current concepts on the diagnosis and management of AHF in order to guide daily clinical practice and to underline the unmet needs. Preventive strategies are also discussed.

## 1. Introduction

AHF is defined as a new onset or recurrence of HF symptoms and signs requiring emergency therapeutic interventions [1,2]. It may occur as the first manifestation of HF, or more frequently as an acute decompensation of chronic HF [3].

Different AHF classification criteria have been proposed, mainly reflecting the clinical heterogeneity of the syndrome [e.g., hemodynamic status (wet/dry–warm/cold) or according to clinical scenario (decompensated heart failure, acute right heart failure, acute pulmonary edema, cardiogenic shock)] [3] (Table 1).

AHF is the most frequent cause of unplanned hospital admissions in patients >65 years of age and is associated with poor outcomes, with in-hospital and 1-year mortality rates of ~10% and ~30%, respectively, with 90-day readmission rates ~20–30% [4]. Moreover, it imposes a significant financial burden to health systems, with the total medical cost of annual median hospitalizations estimated at USD ~16,000 per patient [5,6].

## 2. Epidemiology

The mean age of patients presenting with AHF ranges between 70 and 73 years. About half of patients are male. The majority (65–75%) have a known history of HF. At presentation, most of them have normal or increased blood pressure, while patients presenting with hypotension are generally less than ≤8%, including patients with cardiogenic shock (CS) that represent less than ≤1–2% of cases [7].

Patients presenting with AHF often suffer from several other conditions besides HF. Comorbid states are roughly divided into cardiovascular and non-cardiovascular states. The cardiovascular history usually comprises arterial hypertension (HTN) (~70% of patients), coronary artery disease (CAD) (~50–60%), and atrial fibrillation (AF) (~30–40%) [8,9].

Non-cardiovascular comorbidities include diabetes mellitus (DM) (~40%), renal dysfunction (~20–30%), chronic obstructive pulmonary disease (COPD) (~20–30%), and anemia (~15–30%) [7,10,11].

A significant number of AHF patients (~35–40%) do not have reduced left ventricle ejection fraction (LVEF) [5,12]. In this regard, patients with preserved LVEF are usually older (mean age of 75 years) and more frequently female (~60% of patients). Furthermore, they are less affected by CAD but suffer HTN and DM more frequently [13].

## 3. Management

### 3.1. Pre-Hospital

AHF patients should immediately (‘time-to-treatment’ concept) receive appropriate therapy and be rapidly transferred to the nearest hospital, preferably to a site with an intensive cardiology unit (CCU/ICU) [3,14].

In the pre-hospital setting, AHF patients benefit from non-invasive monitoring, including heart and respiratory rate (HR and RR), BP, pulse oximetry (SpO_2_), and continuous electrocardiogram (ECG) [3,15].

Oxygen therapy should be administered if SpO_2_ < 90%. In patients with respiratory distress, non-invasive ventilation (NIV) should be implemented [3,16].

### 3.2. In Hospital

#### 3.2.1. Triage

AHF patients admitted to the emergency department (ED) with mild symptoms and signs of congestion, no renal dysfunction, negative troponin values, and very low neuropeptide (NP) levels can be discharged directly home after a small dose of diuretics and adjustments of oral therapy as needed. They should be referred to their physician with the advice to be clinically followed by the HF multidisciplinary outpatient clinic [17].

On the other hand, hemodynamically unstable patients should be admitted to the cardiology ward or ICU. In this regard, admission ICU criteria include RR > 25, SpO_2_ < 90%, use of accessory muscles for breathing, SBP < 90 mmHg, need for intubation (or already intubated), or signs of hypoperfusion [oliguria, cold peripheries, altered mental status, lactate > 2 mmol/L, metabolic acidosis, and venous oxygen saturation (SvO_2_) < 65%] [17,18] (Figure 1).

#### 3.2.2. Diagnostic Workup (Figure 2)

##### Step 1

a.Search for reversible causes (Table 2)

Management starts with the search for specific causes of AHF. These include acute coronary syndromes (ACS), hypertensive emergency, rapid arrhythmias or severe bradycardia/conduction disturbances, acute mechanical causes (i.e., acute valve regurgitation), acute pulmonary embolism (PE), infections, and tamponade (CHAMPIT). Dietary and fluid restriction and medication noncompliance should also be ascertained at this time.

After exclusion of these conditions, which need to be treated/corrected urgently, management of AHF differs according to clinical presentation [3,17].

**Figure 2 jcm-12-00846-f002:**
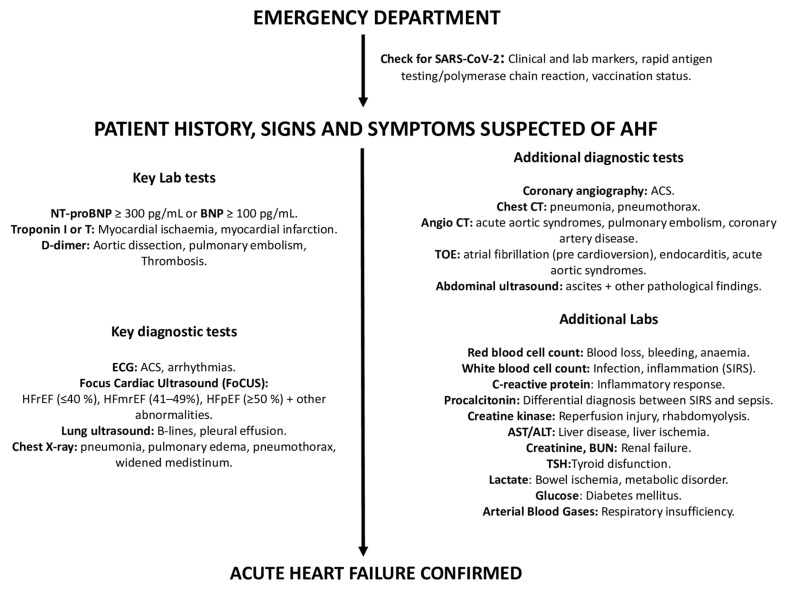
Diagnostic workup of AHF. Abbreviations: ACS: acute coronary syndrome; ALT: alanine aminotransferase; AST: aspartate aminotransferase; BUN: blood urea nitrogen, BNP: brain natriuretic peptide; CT: computed tomography; ECG: electrocardiogram; TOE: transesophageal echocardiogram; TSH: thyroid-stimulating hormone.

**Table 2 jcm-12-00846-t002:** Triggers of AHF.

Triggers	Lab Test	Invasive/Non-Invasive Test	Notes
ACS [19,20]	hs-cTn (I or T)	ECG TTE Coronary angiography	- Immediate primary PCI (or CABG in selected cases) is recommended. - Centers without 24/7 PCI availability must transfer the patient immediately.
Arrhythmias [21,22,23]	Electrolytes, TFTs	ECG TTE Interrogation of ICD (in selected patients)	- Electrical cardioversion is recommended in patients hemodynamically compromised by AF/SMVT and in whom urgent restoration of sinus rhythm is required to improve the patient’s clinical condition rapidly. - ALS/defibrillation in VF/VT without pulse. - Pacing is recommended in patients hemodynamically compromised by severe bradycardia or heart block to improve the patient’s clinical condition.
Acute Myocarditis [24]	hs-cTn (I or T), PCR, ESR, WBC count	ECG + TTE CCT/coronary angiography Endomyocardial biopsy in patients presenting with severe heart failure or cardiogenic shock	- Patients presenting with severe heart failure or cardiogenic shock should immediately be referred to hub centers. - CMRI should be performed within 2/3 weeks from the onset of symptoms when the patient is hemodynamically stable.
Endocarditis [25]	ESR, CRP, blood culture, autoimmunity testing in selected cases	TTE + TOE CCT/total-body CT scan	Patients presenting with severe heart failure or cardiogenic shock should be referred early and managed in a reference center with immediate surgical facilities.
Acute aortic syndromes [26]	D-dimer	TTE + CTA (1st choice) TTE + TOE (2nd choice)	D-dimer is highly sensitive to rule out classical AAD within the first 6 h of symptom onset in low–moderate-risk patients.
Mechanical cause (free wall rupture, ventricular septal defect, acute mitral regurgitation, cardiac tamponade) [19,20]	hs-cTn (I or T), D-dimer	TTE	Prompt intervention/surgery is needed; transfer to Hub center.
Pulmonary embolism [27]	D-dimer, hs-cTn, ABG	ECG + TTE CTPA Compression ultrasonography	If hemodynamically unstable, transfer to ICU.
Hypertension emergency [28]	FBC, creatinine, electrolytes, LDH, haptoglobin, hs-cTn, pregnancy test in women of child-bearing age	Chest X-ray TTE CT or MRI brain in suspected nervous system involvement CTA in suspected acute aortic disease	Patients with severe hypertension associated with AHF require an urgent reduction of BP with IV drug administration.
Pneumonia [29]	FBC, ESR, CRP, PCT	Chest X-ray Chest CT	Admission to an ICU for patients with hypotension requiring vasopressors or respiratory failure requiring mechanical ventilation.
COPD exacerbation or asthma [30]	ABG, PCR, PCT	Chest X-ray Chest CT	Admission to an ICU for patients with hypotension requiring vasopressors or respiratory failure requiring mechanical ventilation.
Thyroid dysfunction [31]	TFTs	ECG TTE	Management of myxedema coma and thyroid storm requires both medical and supportive therapies and should be treated in an ICU setting.
Anemia [32]	FBC	-	Urgent RBC transfusion needed.

Abbreviations: ABG: arterial blood gases; ALS: advanced life support; CABG: coronary artery bypass graft surgery; CCT: cardiac computer tomography; CRP: C-reactive protein; CT: computer tomography; CTA: computed tomography angiography; CTPA: CT pulmonary angiogram; ECG: electrocardiogram; ESR: erythrocyte sedimentation rate; FBC: full blood count; hs-Tn: high-sensitive troponins; ICD: implantable cardioverter defibrillator; ICU: intensive care unit; MRI: magnetic resonance imaging; PCI: percutaneous coronary intervention; PCT: procalcitonin; RBC: red blood cells; TFTs: thyroid function tests; TOE: transesophageal echocardiogram; TTE: transthoracic echocardiogram, SMVT: sustained monomorphic ventricular tachycardia.

b.Check for SARS-CoV-2 infection [33,34]

At the time of hospital admission, it is advisable to:-Search for clinical and laboratory clues suggesting COVID-19 infection;-Perform SARS-CoV-2 rapid antigen testing/polymerase chain reaction;-Check for COVID-19 vaccination status.

c.Assess presenting symptoms and signs

The most common symptoms (reflecting pulmonary and/or systemic congestion) include dyspnea during exercise or at rest, orthopnea, fatigue, and reduced exercise tolerance. Clinical signs usually include peripheral oedema, jugular vein distension, the presence of a third heart sound and pulmonary rales [2].

Symptoms and signs such as cold and clammy skin, altered mental status, and oliguria indicate peripheral hypoperfusion—impending CS [2].

##### Step 2

a.Lab tests

Neuropeptides

Cardiovascular biomarkers play a crucial role in the diagnostic–prognostic process of AHF. Upon presentation to the ED, plasma NP levels (BNP, NT-proBNP, or MR-proANP) should be measured (point-of-care assay) in all patients with acute dyspnea. Due to the strong link with hemodynamic intracardiac stress, they may help to differentiate between cardiac and non-cardiac causes of acute dyspnea [35,36].

Cut-offs for AHF are BNP < 100 pg/mL, NT-proBNP < 300 pg/mL, and MR-proANP < 120 pg/mL, with normal NP concentrations making the diagnosis of AHF extremely unlikely.

However, there are many causes of elevated NP levels—both cardiovascular (CV) and non-CV—that might reduce their diagnostic accuracy. These causes include AF, increasing age, and acute or chronic kidney disease. Conversely, NP concentrations may be disproportionately low in obese patients, in patients with pre-left ventricle causes of HF (i.e., mitral stenosis and acute mitral regurgitation), or pericardial diseases.

As a note, NT-pro BNP instead of BNP should be tested in patients taking sacubitril-valsartan [37].

It should also be highlighted that NP levels are strong predictors of readmissions and death [38].

Troponin

In addition to ACS, elevated high-sensitivity troponin I/T (hsTn I/T) levels may be observed in most non-ACS AHF patients and are associated with worse in-hospital and post-discharge outcomes [39].

Others

Further lab tests (i.e., BUN (or urea), creatinine, electrolytes, glucose, complete blood count, procalcitonin, PCR, and D-dimer) may be useful to detect and/or to confirm clinically suspected comorbidities and/or end-organ damage [15].

SpO_2_/arterial blood gas (ABG)

SpO_2_ should be measured routinely at the time of AHF patient presentation and continuous monitoring may be needed in the first hours or days.

Routine ABG is not needed. Specific indications for ABG are: respiratory distress [defined as acute increase in the work of breathing or significant tachypnea (RR > 25 breaths/min)], documented hypoxemia (SpO_2_ < 90%) not responsive to supplemental oxygen, and evidence of acidosis or elevated lactate levels. In the case of respiratory failure, ABG may show PaO_2_ < 60 mmHg, PaCO_2_ > 45 mmHg or PaO_2_/FiO_2_ < 300 mmHg. Of note, venous sample might acceptably indicate pH and CO_2_ [15].

b.ECG

Routine admission ECG is recommended since it can exclude ACS and arrhythmias. In this regard, careful attention should be paid to ECG changes suggestive of myocardial ischemia. Tachyarrhythmias [i.e., AF (present in 20% to 30% patients), ventricular tachycardia] or bradyarrhythmias (i.e., advanced atrio-ventricular blocks) are also a common trigger for AHF [3,21,22].

c.Chest X-ray

Chest X-ray may reveal lung congestion and/or pleural effusion. Furthermore, it may identify non-cardiac-disease causes of the patient’s symptoms (i.e., pneumonia, pneumothorax, widened mediastinum).

d.Transthoracic echocardiography (TTE)

TTE represents the single most useful imaging technique to investigate AHF etiology and to guide related therapeutic interventions.

A “Focus Cardiac Ultrasound” (FoCUS), followed by comprehensive TTE exam, is recommended in all patients to assess LV global systolic (reduced vs. preserved EF) and diastolic function, regional wall abnormalities, valvular heart (stenosis and/or regurgitations) and pericardial disease. In addition, it is of paramount importance to evaluate right heart structure and function, as well as pulmonary pressures, as these are major prognostic determinants [40].

As a note, an E:E’ ratio greater than 15 predicts a pulmonary arterial wedge pressure (PAWP) greater than 15 mm Hg, and has been demonstrated to be accurate in the ED and intensive care settings [2,41] (Figure 3).

e.Lung ultrasound (LUS)

LUS has emerged as a valuable modality to detect and monitor pulmonary congestion in patients with AHF in a low-cost, portable, real-time, and radiation-free manner.

It outperforms the diagnostic accuracy of the chest radiograph in the detection of pleural water (pleural effusion) and lung water (pulmonary congestion as multiple B-lines) [42].

B-lines are well defined (laser-like), hyperechoic, vertical comet-tail artifacts that arise strictly from the pleural line, move in sync with lung sliding and spread to the edge of the screen without fading and erasing A lines. The number of lines is proportional to the severity of congestion and identifies the cardiogenic origin of dyspnea with 85% sensitivity and 92% specificity [43].

The B profile is useful to track dynamic changes in pulmonary congestion in responses to treatment, and its persistence at pre-discharge or in clinically stable outpatients with heart failure is predictive of heart failure hospitalization or death [44].

The amount of pleural effusion can be scored as trivial (<2 mm), small (2 to 15 mm), moderate (15 to 25 mm), or large (>25 mm). Furthermore, LUS represents a guide to thoracentesis in patients with AHF and at least moderate pleural effusion [45].

As a note, the evaluation of “lung sliding” (a horizontal, to-and-fro movement, beginning at the pleural line and synchronous with respiration) is helpful in the differential diagnosis of several parenchymal lung diseases that are present as comorbidities in HF or as causes of dyspnea suspected to be cardiac in origin. For instance, “lung sliding” disappears in pneumothorax and it is reduced or abolished in the case of pneumonia, acute respiratory distress syndrome (ARDS), or pleural adhesions [43].

f.Abdominal ultrasound (AUS)

AUS can be useful for measurement of the inferior vena cava (IVC) diameter as an indirect measure of right atrial pressures (IVC < 21 mm that collapses >50% suggests normal right atrial pressure) [46].

In HF patients, an increased IVC diameter might detect abnormal intravascular volume even prior to any change in symptoms or body weight, and in turn monitor the response to diuretics. AUS can also detect ascites and abdominal aortic aneurysm [46].

Recently, ultrasound techniques have also been implemented to assess renal blood flow [47].

g.Transesophageal echocardiogram (TEE)

TEE may be performed in suspected endocarditis and acute aortic syndromes (AAS). Furthermore, it may be useful to better define heart valve abnormalities and to detect intracardiac shunt and thrombi. Absolute contraindications include: unrepaired tracheoesophageal fistula, esophageal obstruction/stricture, perforated hollow viscus, active gastric/esophageal bleeding, poor airway control, severe respiratory depression, and uncooperative, unsedated patient [48].

##### Step 3. Additional Non-Invasive and Invasive Tests

a.High-resolution chest computed tomography (Chest HR-CT)

Chest HR-CT should be considered when pulmonary parenchymal component is suspected among patients presenting with AHF.

CT can also identify signs of pulmonary edema, such as interlobular septal thickening, fissural thickening, peribronchovascular thickening, perihilar or bat-wing appearance of oedema, increased artery-to-bronchus ratio, pleural effusion, and cardiac enlargement in more advanced HF [49,50].

Furthermore, high-resolution CT provides an effective modality to evaluate patients with suspected COVID-19.

b.Chest CT angiography (CTA)

CTA can be used as a one-step imaging modality (dual rule-out strategy) to exclude PE or AAS. It can be performed with most CT equipment. Furthermore, with state-of-the-art CT equipment, synchronizing image acquisition with the cardiac cycle, it is possible to perform the so-called Triple Rule-Out strategy (TRO). This protocol allows the heart and the coronary arteries to be imaged, allowing the exclusion of ACS in a clinical context where this diagnosis might not be straightforward. The main drawbacks of CTA are the administration of iodinated contrast agent, which may cause acute kidney injury or allergic reactions, even though the amount of contrast material currently required to perform the scan is quite low compared to in the past (i.e., using state-of-the-art CT technology, 50 mL). Furthermore, the use of ionizing radiation should be avoided in younger patients, especially women [51].

Recent CT technology also allows the performance of a full anatomical and functional assessment of cardiac and thoracic structures. Hence, a patient undergoing this kind of assessment will have all heart chamber volumes and functionality assessed, the presence of thrombosis within the cardiac chambers ruled out, the superior and inferior vena cava assessed for patency and distention, the pulmonary artery evaluated for dilatation, and so forth. When COVID-19 is assessed in the context of a TRO protocol, it is referred to as Quadruple Rule-Out [52]. When other causes for the acute settings are included in the evaluation, it can be referred to as Quintuple Rule-Out. Because of this flexibility and wide range of rule-in/rule-out capabilities and its relatively easy access, CT is already, and will become, an increasingly central tool in all acute clinical settings (Figure 4 and Figure 5).

CIN (contrast induced nephropathy) remains one of the most serious complications of iodinated contrast medium (CM). It is defined as a ≥25% increase in serum creatinine from the baseline value, or an absolute increase of at least 0.5 mg/dL (44.2 µmol/L), 48–72 h after the administration of radiographic contrast media that is not attributable to other causes [53].

Pre-existing renal impairment represents the most important risk factor for CIN. The baseline renal function of patients undergoing contrast studies is best assessed with calculations of glomerular filtration rate (GFR), such as the MDRD or Cockcroft–Gault formulae in adults [53].

Patients at high risk of developing CIN should be identified early and prophylactic measures implemented before the procedure (Table 3).

The frequency of allergic-like adverse events related to the intravascular administration of iodinated CM is low and has decreased considerably since the use of nonionic low-osmolality contrast media. However, the majority of adverse side effects to CM are mild non-life-threatening events that usually require only observation, reassurance, and/or supportive measures [54]. Severe reactions (i.e., bronchospasm, laryngeal edema, anaphylaxis) occur rarely and are unpredictable. A frequently recommended premedication oral regimen for elective examinations is shown in Table 4.

c.Coronary angiography

In AHF patients with a clinical picture related to ACS, an immediate coronary angiography, along with revascularization (if needed), should be performed [19,20].

## 4. In-Hospital Therapeutic Interventions

The main goals of treatment in AHF consist of alleviating symptoms, improving congestion and organ perfusion, restoring oxygenation, and preventing thromboembolism.

### 4.1. Pharmacologic

#### 4.1.1. Diuretics (Table 5)

The cornerstone of AHF treatment is represented by diuretics with IV loop diuretics (e.g., furosemide, bumetanide or torasemide) used as first-line therapy in patients with AHF and congestion [3].

**Table 5 jcm-12-00846-t005:** Diuretics [2,55].

Drug	Mechanism of Action	Dose	Adverse Reactions	Notes
Diuretics				
Used in hypervolemia to relief symptoms of congestion				
Loop diuretics				
Furosemide *, Torsemide *, Bumetanide.	Sulfonamide loop diuretics. Inhibit cotransport system (Na^+^/K^+^/2Cl^−^) of thick ascending limb of loop of Henle. Abolish hypertonicity of medulla, preventing concentration of urine. Associated with increased PGE (vasodilatory effect on afferent arteriole). Increase Ca^2+^ excretion.	Initial dose, diuretic-naive: - furosemide: 20–40 mg IV - torsemide: 10–20 mg IV - bumetanide: 0.5–1 mg IV Initial dose, for those on chronic diuretics: 1–2 times the daily oral chronic dose as intermittent IV boluses or continuous IV infusion. Adjust dose to relieve symptoms, reduce volume excess, and avoid hypotension.	Ototoxicity, hypokalemia, hypomagnesemia, dehydration, allergy, metabolic alkalosis, nephritis, gout.	Monitor symptoms, urine output, renal function, and serum electrolytes regularly during therapy. Consider continuous infusion in diuretic-resistant patients. A satisfactory diuretic response can be defined as a urine sodium content >50–70 mEq/L at 2 h and/or by a urine output >100–150 mL/h during the first 6 h.
Thiazide diuretics				
Hydrochlorothiazide *, chlorthalidone, metolazone.	Inhibit NaCl reabsorption in early distal convolute tubule. Decrease Ca^2+^ excretion.	- Hydrochrothiazide: start with 25 mg PO once or twice daily (dose range: 12.5–200 mg/day) - Chlorthalidone: start with 25 mg PO once daily (dose range: 12.5–200 mg/day) - Metolazone: start with 1.25–5 mg PO 1–7 times/week (dose range: 1.25–20 mg/day)	Hypokalemic metabolic alkalosis, hyponatremia, hyperglycemia, hyperlipidemia, hyperuricemia, hypercalcemia. Sulfa allergy.	Use with caution in patients with severe renal disease, hepatic impairment, or progressive liver disease.
Potassium-sparing diuretics				
Spironolactone *, Eplerenone *, Amiloride, Triamterene.	Spironolactone and eplerenone are competitive aldosterone receptor antagonists in cortical collecting tubule. Amiloride blocks Na+ channels at the same part of the tubule.	- Spironolactone: start with 12.5–25 mg PO daily (target dose: 25–50 mg PO daily) - Eplerenone: start with 25 mg PO once daily (target dose: 50 mg PO once daily) - Amiloride: start with 5 mg PO once daily (dose range: 1.25–20 mg/day).	Hyperkalemia (can lead to arrhythmias), endocrine effects with spironolactone (e.g., gynecomastia, antiandrogen effects).	Monitor serum potassium.

*: Principal drugs. Abbreviations: IV: intravenous; PGE: prostaglandin E; PO: per os.

The use of an IV dose of diuretics at least equal to the pre-existing oral dose is recommended in those already receiving oral diuretics, and 20–40 mg IV furosemide (or equivalent) in those who are not on regular oral diuretics [3,56].

Furosemide can be given as 2–3 daily boluses or as a continuous infusion. Daily single bolus administrations are discouraged for the possibility of post-dosing sodium retention [3,56].

The diuretic response is evaluated by measuring the urinary volume output and/or spot urinary sodium content, with a satisfactory diuretic response defined as a urine sodium content >50–70 mEq/L at 2 h and/or by a urine output >100–150 mL/h during the first 6 h [56].

If there is an insufficient diuretic response, the loop diuretic IV dose can be doubled. Transition to oral treatment should be started when the patient’s clinical condition is stable.

In patients with resistant oedema, dual treatment with a loop diuretic and a thiazide or a thiazide-like diuretic (e.g., metolazone) may be considered to achieve adequate diuresis (so-called “sequential nephron blockade”) [56].

#### 4.1.2. Vasodilators (Table 6)

Intravenous vasodilators may be considered to relieve AHF symptoms when SBP is >110 mmHg [3].

They may be started at low doses and up-titrated to achieve clinical improvement and BP control. Nitrates are generally administered with an initial bolus followed by continuous infusion. However, these agents should be avoided in patients with concurrent obstructive valvular disease (i.e., severe aortic stenosis) or restrictive physiology (i.e., hypertrophic cardiomyopathy) [57].

**Table 6 jcm-12-00846-t006:** Vasodilators [2,57].

Drug	Mechanism of action	Dose	Adverse reactions	Notes
Vasodilators				
Used for relief of dyspnea in patients without hypotension (SBP > 110 mmHg), potentially useful in severely congested patients with hypertension or severe mitral valve regurgitation complicating LV dysfunction.				
Nitroglycerine Isosorbide dinitrate	Vasodilate by increasing NO in vascular smooth muscle that leads to increase of cGMP and smooth muscle relaxation (veins > arteries).	Nitroglycerine: start with 10–20 μg/min, increase up to 200 μg/min IV. Isosorbide dinitrate: start with 1 mg/h, increase up to 10 mg/h IV.	Hypotension, reflex tachycardia, headache. Tolerance in continuous use.	Contraindicated in right ventricular infarction, hypertrophic cardiomyopathy, severe aortic stenosis and with concurrent PDE-5 inhibitor use.
Nitroprusside	Short acting vasodilator (arteries = veins). Increases cGMP via direct release of NO.	Start with 0.3 μg/kg/min and increase up to 5 μg/kg/min IV.	Hypotension, isocyanate toxicity, light sensitivity.	Contraindicated in right ventricular infarction, hypertrophic cardiomyopathy, severe aortic stenosis, and with concurrent PDE-5 inhibitor use.

Abbreviations: cGMP: cyclic guanosine monophosphate; IV: intravenous; NO: nitric oxide; PDE: phosphodiesterase; SBP: systolic blood pressure.

#### 4.1.3. Opiates

Although the routine use of opiates (i.e., morphine) in AHF is not recommended, they may be considered in selected patients, particularly in case of severe pain, anxiety or in the setting of palliation [58,59].

#### 4.1.4. Digoxin

Digoxin is mostly indicated (boluses of 0.25–0.5 mg IV if not used previously, followed by an oral or IV dose of 0.25 mg at least 12 h after the initial dose) in patients with AF and rapid ventricular rate (>110 bpm) despite beta-blockers [3,60].

Caution should be taken in the elderly or in patients with factors affecting digoxin metabolism (i.e., renal failure, drug interaction) [3].

Furthermore, unless the risk of toxicity outweighs the benefit, discontinuation of digoxin is generally discouraged. In this regard, an association between withdrawal of therapy and worsening HF has been well documented [60].

#### 4.1.5. Anticoagulants

AHF patients are at high risk of deep venous thrombosis (DVT) and PE as a direct consequence of higher venous pressures and lower cardiac output. In this regard, current guidelines support the use of thromboprophylaxis [e.g., low-molecular-weight heparin (LMWH) given at 4000 to 5000 units daily, or 2500 to 3000 units twice daily subcutaneously] in all appropriate hospitalized AHF patients, unless contraindicated [61].

In addition, oral anticoagulation [preferring new oral anticoagulants (NOACs) to vitamin K antagonists (VKAs), except in patients with mechanical heart valves or moderate–severe mitral stenosis] is recommended in AHF patients with paroxysmal, persistent, or permanent AF with a CHA2DS2-VASc score ≥ 2 in men and ≥ 3 in women. The HAS-BLED score should be considered to identify patients at high risk of bleeding (HAS-BLED score ≥ 3) for early and more frequent clinical assessments and follow-up [22].

#### 4.1.6. Inotropes/Vasopressors (Table 7)

Inotropes [including sympathomimetics/synthetic catecholamines (e.g., dobutamine, adrenaline), phosphodiesterase inhibitors (e.g., milrinone, enoximone), and, more recently, Ca^2+^ sensitizers (e.g., levosimendan)] should be reserved for patients with LV systolic dysfunction, low cardiac output and low SBP (e.g., <90 mmHg), resulting in poor vital organ perfusion [2].

Inotropes improve myocardial contractility, but, especially in the case of the sympathomimetics, also increase myocardial O_2_ consumption. As a direct consequence they may trigger supraventricular and ventricular tachyarrhythmias. In this regard, it should be underlined that all patients under inotrope treatment require close monitoring of cardiac rhythm and hemodynamic parameters [62,63].

**Table 7 jcm-12-00846-t007:** Inotropes/vasopressors [2,63].

Drug	Mechanism of Action	Dose	Adverse Reactions	Notes
Inotropes/Vasopressors				
Used for maintenance of systemic perfusion and preservation of end organ function in patients with severe systolic dysfunction presenting with hypotension (<90 mmHg) or low cardiac output in the presence of congestion and organ hypoperfusion.				
Dobutamine	Agonist of both beta1- and beta2-adrenergic receptors with variable effects on the alpha receptors	Continuous IV infusion rate of 2–20 mcg/kg/minute	Hypotension, increased myocardial oxygen demand, phlebitis	Continuously monitor ECG and blood pressure. Dobutamine is preferred over milrinone in patients who are acutely unstable or hypotensive, or those with renal insufficiency.
Dopamine	Agonist of both adrenergic and dopaminergic receptors	Infusion rate of 3–5 μg/kg/min; inotropic (beta+); >5 μg/kg/min: (beta+), vasopressor (alpha+)	Arrhythmias, tachycardia	Continuously monitor ECG and blood pressure. Clinical effects are dose-related; low doses increase renal blood flow/urine output, intermediate doses also increase cardiac contractility and chronotropy, and high doses result in vasoconstriction.
Milrinone	PDE inhibitor (increases cAMP)	Bolus: 25–75 μg/kg over 10–20 min then infusion rate of 0.375–0.75 μg/kg/min continuous IV infusion.	Tachycardia, ventricular arrhythmias, hypotension	Continuously monitor ECG and blood pressure. Not recommended in acutely worsened ischemic heart failure.
Levosimendan	Cardiac Ca^2+^ channels sensitizer. Activator of K^+^ channels of vascular smooth muscle cells.	Optional bolus: 2 μg/kg over 10 min, infusion rate of 0.1 μg/kg/min, which can be decreased to 0.05 or increased to 0.2 μg/kg/min.	Tachycardia, ventricular arrhythmias, hypotension.	Continuously monitor ECG and blood pressure. Bolus not recommended in hypotensive patients.
Norepinephrine	Potent agonist of the beta1 and the alpha 1 receptors	Infusion rate of 0.2–1.0 μg/kg/min.	End-organ hypoperfusion and tissue necrosis, arrhythmias.	Continuously monitor ECG and blood pressure.
Epinephrine	Full beta receptor agonist	Infusion rate of 0.05–0.5 μg/kg/min. A bolus of 1 mg can be given IV during resuscitation, repeated every 3–5 min.	End-organ hypoperfusion and tissue necrosis, arrhythmias.	Continuously monitor ECG and blood pressure. Use should be restricted to patients with persistent hypotension despite adequate cardiac filling pressures and the use of other vasoactive agents, as well as for resuscitation protocols.

Abbreviations: cAMP: cyclic adenosine monophosphate; ECG: electrocardiogram; IV: intravenous; PDE: phosphodiesterase.

Of note, while inotropes have been shown to improve symptoms and signs of congestion, these agents have failed to reveal any improvement in mortality in patients with AHF [64].

#### 4.1.7. Future Directions

In the EMPULSE trial, early initiation of SGLT-2 inhibitor empagliflozin in patients hospitalized for AHF led to a statistically significant clinical benefit at 90 days with fewer deaths, improvement in quality of life, lower NT-pro BNP levels, and weight loss [65,66].

The ADVOR trial has reported that, when used in combination with loop diuretic, acetazolamide (a carbonic anhydrase inhibitor) can lead to a greater incidence of successful decongestion [67].

Istaroxime, a novel compound with inotropic and lusitropic positive properties and a dual mechanism of action (activation of the sarcoplasmic reticulum Ca^2+^/ATPase 2a (SERCA2a) and inhibition of the Na^+^/K^+^-ATPase), has been shown to increase SBP without activating the adrenergic system, and to improve pulmonary capillary wedge pressure and diastolic cardiac function [68,69,70,71].

Furthermore, in AHF patients, early administration (within 16 h) of serelaxin, a peptide involved in cardiovascular adaptations during pregnancy, has been shown to be associated with a reduction in 5-day worsening HF and markers of renal dysfunction [72].

#### 4.1.8. Management of Chronic HF Therapy

Temporary discontinuation of angiotensin-converting enzyme (ACE), inhibitor/angiotensin receptor blockers (ARB), or beta-blockers may be necessary in the settings of CS or symptomatic hypotension. ACE-I/ARB and mineralocorticoid receptor antagonists (MRAs) may also need to be temporarily held in case of renal dysfunction, oliguria, and/or hyperkalemia [73].

The Initiation of beta-blocker therapy during AHF is contraindicated due to acute negative inotropic effects. However, initiation of beta-blocker in euvolemic patients prior to discharge is safe and associated with increased long-term survival [74].

### 4.2. Non-Pharmacologic

#### 4.2.1. Mechanical Ventilation

NIV consists of applying positive intrathoracic pressure (PIP) to conscious patients through different interfaces, and can be either continuous positive airway pressure (CPAP) or bilevel positive airway pressure (BiPAP) [75].

It should be highlighted that NIV has to be started as soon as possible in patients with respiratory distress (respiratory rate >25 breaths/minute, SpO_2_ < 90%) to improve gas exchange and reduce the rate of endotracheal intubation [3].

Absolute contraindications to NIV include [75]:Cardiac or respiratory arrest;Anatomical abnormality (unable to fit the interface);Inability to keep patent airway (uncontrolled agitation, coma or obtunded mental status);Refractory hypotension.

If there is only hypoxemia, CPAP is the treatment of choice. In cases of hypoxemia and hypercapnia, BiPAP is preferred. CPAP is generally started at a pressure of 5 cm H_2_O, which is increased in a stepwise manner to up to 10 cm H_2_O. In BiPAP, it is reasonable to start with an EPAP of 5 cm H_2_O and an IPAP of 10–14 cm H_2_O. EPAP and IPAP can be adjusted further according to the effect on oxygenation and ventilation, respectively [75].

The response to NIV should be assessed after 60 min, and thereafter on a continuous basis. Signs of NIV failure are patient fatigue, progressive worsening of level of consciousness, hemodynamic instability, persistent tachypnoea (>35 breaths/minute), and progressive worsening of respiratory failure with acidosis, hypoxemia, or hypercapnia [75].

Endotracheal intubation and mechanical ventilation are only required in a minority of AHF patients, as most of them will respond to NIV.

Criteria for endotracheal intubation are the following [75]:Cardiac or respiratory arrest;Progressive worsening of altered mental status;Progressive worsening of pH, PaCO_2_, or PaO_2_ despite NIV;Progressive signs of fatigue during NIV;Need to protect the airway;Persistent hemodynamic instability;Agitation or intolerance to NIV with progressive respiratory failure.

#### 4.2.2. Electric Cardioversion

AF patients presenting with a rapid ventricular rate and acute hemodynamic instability (i.e., acute pulmonary oedema, ongoing myocardial ischemia, symptomatic hypotension or CS) require prompt intervention, and emergency electrical cardioversion should be attempted without delay. In this setting, amiodarone may also be considered in order to control heart rate response [3,22].

#### 4.2.3. Mechanical Circulatory Support (MCS)

Short-term MCS (which increases cardiac output and supports end organ damage) may be implemented as a bridge to recovery (BTR), bridge to decision (BTD) or bridge to transplant (BTT) (Table 8). Intra-aortic balloon pump (IABP) is not routinely recommended [76].

#### 4.2.4. Renal Replacement Therapy

Ultrafiltration (i.e., hemodialysis) may be indicated in case of refractory congestion non-responsive to diuretics [17]. It may be considered if the following criteria are met:Oliguria unresponsive to fluid resuscitation measures;Severe hyperkalemia (K^+^ > 6.5 mmol/L);Severe acidemia (pH < 7.2);Serum urea level > 25 mmol/L (> 150 mg/dL);Serum creatinine > 300 mmol/L (> 3.4 mg/dL) that is worsening.

## 5. Daily Patient Monitoring

Daily patient monitoring includes:Weight check along with completion of an accurate fluid balance chart;Standard non-invasive monitoring of HR, RR, BP;Renal function and electrolyte measurement.

Invasive monitoring with pulmonary artery catheter failed to show any positive influence on inpatient or follow-up outcomes of patients admitted with AHF, and should be carefully used for selected patients [77].

## 6. Pre-Discharge and Post-Discharge Planning

### 6.1. Pre-Discharge

Once hemodynamic stabilization is achieved with IV therapy, treatment should be optimized before discharge according to current HF guidelines in order to (a) relieve congestion, (b) treat comorbidities, and (c) initiate or restart oral optical medical treatment (OMT) [3,78,79,80,81,82].

Indicators of good response to initial therapy that might be considered in discharge include [15]:

Patient-reported subjective improvement;Resting HR < 100 bpmLack of orthostatic changes in BP;Adequate urine output;SpO_2_ > 95% in room air;Decreased body weight.

### 6.2. Post-Discharge (Figure 6)

In order to reduce hospitalizations and mortality, enrollment in a HF multidisciplinary management program is recommended as it has been shown to improve outcomes based on three main aspects [3,14]:Patient self-monitoring (i.e., regular weight checks, adherence to therapy, structured exercise program, and dietary sodium and fluid restriction).Periodic follow-up visits, including monitoring of signs and symptoms of HF, assessment of volume status, BP, HR, and laboratory tests primarily of renal function, electrolytes, iron status, hepatic function, and NP. In patients with minimal symptoms of HF, comparison of NP level with predischarge values should be considered to detect worsening subclinical congestion. At the visit, the physician should also verify that the patient is receiving all guideline-directed chronic HF therapies for which they are eligible. Likewise, laboratory monitoring for corresponding drug adverse effects (i.e., renal insufficiency, electrolyte disturbances) should be considered [3]. Furthermore, planning for additional diagnostic and interventional procedures can be undertaken, including device therapy. It should be highlighted that the 2021 European Society of Cardiology (ESC) HF guidelines recommend the first follow-up outpatient visit within 1 to 2 weeks after discharge [83].Remote monitoring via telemedicine/teleconsulting evaluations. Home telemonitoring can help maintain quality of care, facilitate rapid access to care when needed, reduce patient travel costs, and minimize the frequency of clinic visits [84]. Remote pulmonary arterial pressure monitoring with implantable pressure sensors, with adjustment of diuretic therapy according to pulmonary arterial pressure measurements, substantially reduced HF hospitalizations and improved outcomes in both patients with HFpEF and HFrEF [85].

**Figure 6 jcm-12-00846-f006:**
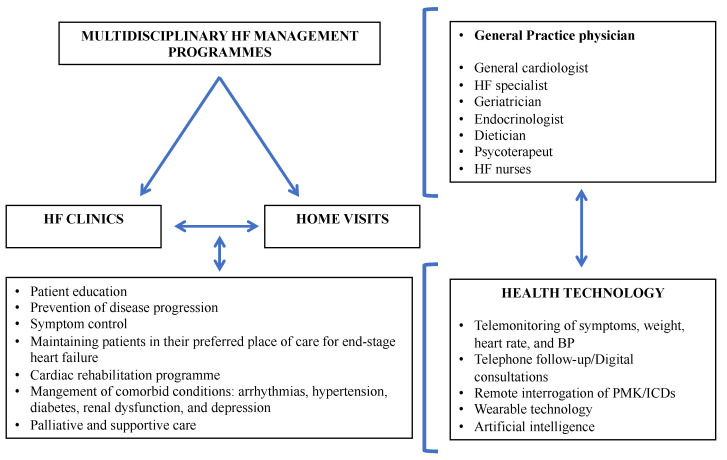
Outpatient management. Abbreviations: BP: blood pressure; HF: heart failure; ICD: implantable cardiac defibrillator; PMK: pacemaker.

## 7. In-Hospital and Long-Term Outcomes

### 7.1. In-Hospital Outcomes

AHF is characterized by relatively low in-hospital mortality but a high rate of recurrent post-discharge events. AHF inpatient mortality ranges between 3% and 7%, with the exception of patients with CS, who have an in-hospital mortality of approximately 40% [5].

At hospital admission, specific predictors of poor prognosis consist of advanced age, HF hospitalization history, decreased kidney function, high NP concentrations, and low BP.

Furthermore, higher degrees of congestion are associated with longer hospital stay [38].

Persistent congestion and high NP levels at discharge are predictors of worse quality of life, recurrent rehospitalization, and higher mortality [86].

### 7.2. Long-Term Outcomes

Approximately 25% of patients hospitalized with HF are readmitted within 30 days of discharge, and mortality during this period can approach 10%. Rates of rehospitalization within 6 months approach 50% in many cohorts, particularly the elderly [83].

In the EVEREST trial, careful adjudication of post-discharge hospitalizations showed that 46% were for HF, 15% for other CV causes, and 39% for non-CV causes.

Of note, approximately half of rehospitalizations are not HF-related, which underscores the high burden of comorbidity as well as the challenges of implementing personalized therapeutic interventions [87].

Median survival in HF patients decreases gradually with the number of hospitalizations, ranging from 2.5 years in patients with one hospital admission to 0.5 years in those with four admissions [88].

## 8. Preventive Strategies

The lifetime risk of HF is approximately 20%, and the prevalence and burden of HF will likely continue to increase in developed countries [77].

In all patients, the cornerstone should be counseling on the importance of healthy lifestyle to optimize CV health [89,90].

In this regard, it is essential to assess modifiable HF risk factors, including HTN, elevated body mass index (BMI), physical inactivity, DM, CAD, and tobacco and alcohol use. It should be highlighted that controlling HTN is associated with a lower risk of incident HF, with current guidelines recommending targeting BP < 130/80 mmHg [90].

Furthermore, in patients with DM, a target HbA1c < 7.0% (53 mmol/mol) is recommended [90].

It is recommended that all patients with HF are regularly screened for anemia and iron deficiency with full blood count, serum ferritin concentration, and transferrin saturation (TSAT). In patients with HF, iron deficiency is defined as either a serum ferritin concentration < 100 ng/mL or 100–299 ng/mL with TSAT < 20% [91].

Ion supplementation with IV ferric carboxymaltose should be considered for the improvement of symptoms, exercise capacity, and quality of life in patients with HF and LVEF < 45%. It should also be considered for the reduction of HF rehospitalizations in patients with LVEF < 50% recently hospitalized for worsening HF [3].

Influenza and pneumococcal vaccination, as well as COVID-19 vaccination, when available, should be considered in patients with HF [92].

## 9. Conclusions

AHF is a life-threatening medical emergency requiring immediate therapeutic interventions in order to optimize hemodynamic status. Precipitants and comorbid conditions should be addressed, specifically acute decompensation triggers such as ACS, hypertensive emergency and malignant arrythmia. Multidisciplinary comprehensive follow-up and rehabilitation programs are recommended, along with the implementation of digital health (i.e., remote monitoring, teleconsulting, and implantable device interrogation) in order to reduce the risk of recurrent HF hospitalization and mortality. In the near future, we may expect a major practical change towards personalized care.

## Figures and Tables

**Figure 1 jcm-12-00846-f001:**
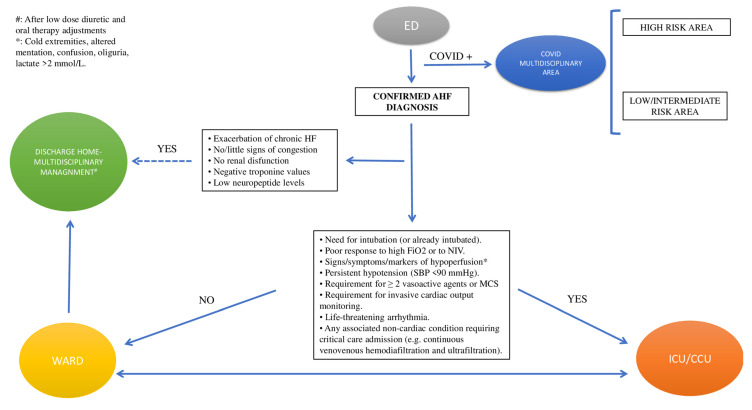
Triage. Abbreviations: AHF: acute heart failure; ED: emergency department; ICU/CCU: intensive cardiology unit/critical care unit; MCS: mechanical circulatory support; NIV: non-invasive ventilation; SBP: systolic blood pressure.

**Figure 3 jcm-12-00846-f003:**
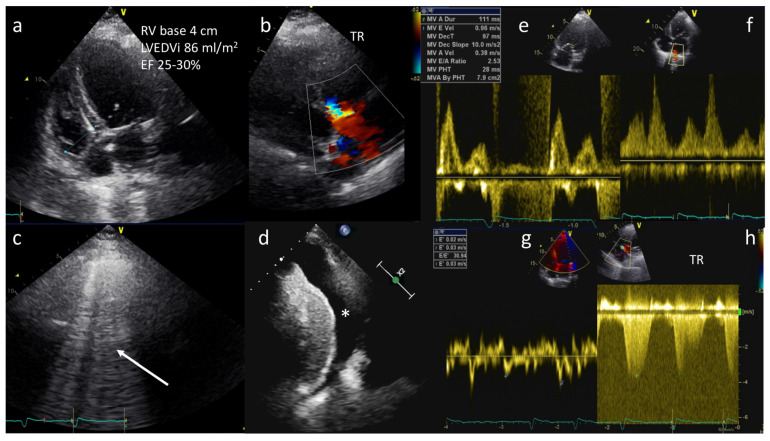
A 68-year-old female with history of dilated cardiomyopathy was admitted for shortness of breath, fatigue, and low-extremity edema. A diagnosis of acute pulmonary edema was made. TTE showed severe LV dilation (LVEDVi 86 mL/m^2^), severe reduction in ejection fraction (EF 25–30%), mildly dilated right ventricle (basal diameter 4 cm) (**a**), and moderate tricuspid regurgitation (TR) (**b**). Lung ultrasound showing B-lines in all sites explored (arrow) and pleural effusion (*) (**c**,**d**). Diastolic dysfunction with increased left ventricular end-diastolic pressures (LVEDP) and PAWP (E/A: 2.5, E/E’: 30 and S < D on pulmonary veins) (**e**–**g**) and estimated pulmonary artery pressure of 70 mmHg (**h**).

**Figure 4 jcm-12-00846-f004:**
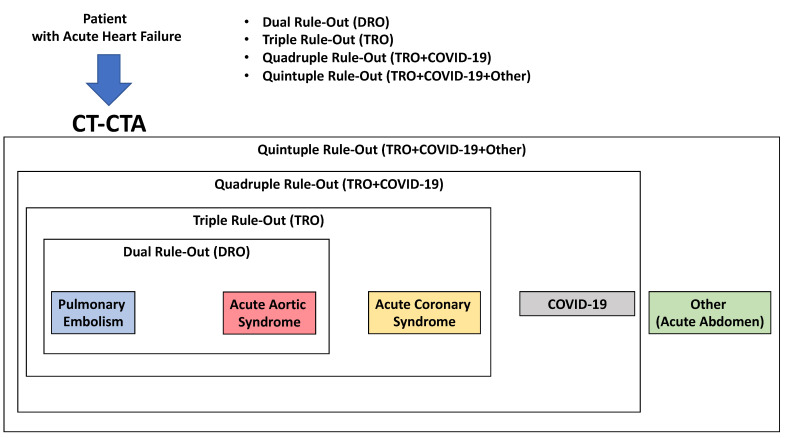
Use of CT in acute heart failure.

**Figure 5 jcm-12-00846-f005:**
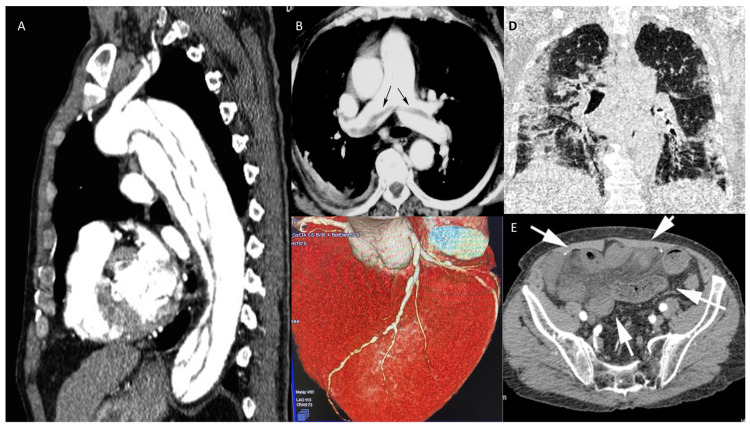
CT in different scenarios. Type B thoracic aortic dissection: post-contrast sagittal CT reconstruction of the aorta demonstrates a medio-intimal flap that begins below left subclavian arterial origin and extends up to diaphragmatic hiatus (**A**). Pulmonary thromboembolism: post-contrast axial CT reconstruction depicts linear contrast defects inside the lumen of main pulmonary arteries (arrows) due to thromboembolism (**B**). Volume rendering post-contrast CT of the left descending coronary artery depicts a brief stenosis of the medium segment (**C**). Coronal unenhanced chest CT shows ground glass opacities of the lungs, especially on the left side, due to interstitial COVID pneumonia (**D**). Post-contrast axial CT image of the pelvis demonstrates ileal loops ischemia with a stratified appearance of the ileal loop’s wall (arrows) due to intramural edema and low submucosal enhancement associated with mesenterial free fluid. (**E**).

**Table 1 jcm-12-00846-t001:** Classification of AHF.

	Acutely Decompensated Heart Failure	Acute Pulmonary Oedema	Isolated Right Ventricular Failure	Cardiogenic Shock
Description	Progressive fluid retention in patients with history of HF	Lung congestion and acute respiratory failure	RV dysfunction and/or pre-capillary pulmonary hypertension	Severe cardiac dysfunction with marked hypotension (SBP < 90 mmHg) despite adequate LV filling pressure
Onset	Gradual (days)	Rapid (hours)	Gradual/rapid	Gradual/rapid
Main clinical presentation	Wet and warm (rarely wet and cold)	Wet and warm (rarely wet and cold)	Wet and cold	Wet and cold
Heart rate	↑	↑	Usually ↓	↑
SBP	Variable	Variable	↓	↓
Cardiac index	Variable	Variable	↓	↓
Hypoperfusion	+/−	+/−	+	+
PCWP	↑↑	↑↑↑	↓	↑↑
Main treatment	Diuretics Inotropic agents/vasopressors (If peripheral hypoperfusion/hypotension) Short-term MCS or RRT if needed	O_2_ (CPAP/NIV) Diuretics Vasodilators Inotropic agents/vasopressors (If peripheral hypoperfusion/hypotension) Short-term MCS or RRT if needed	Diuretics for congestion Inotropic agents/vasopressors (If peripheral hypoperfusion/hypotension) Short-term MCS or RRT if needed	Inotropic agents/vasopressors Short-term MCS or RRT if needed

Abbreviations: CPAP: continuous positive airway pressure; HF: heart failure; LV: left ventricle; MCS: mechanical circulatory support; NIV: non-invasive ventilation; PCWP: pulmonary capillary wedge pressure; RRT: renal replacement therapy; RV: right ventricle; SBP: systolic blood pressure; ↑: increase; ↓: decrease. Modified from “McDonagh T.A.; et al.; 2021 ESC Guidelines for the diagnosis and treatment of acute and chronic heart failure” [3].

**Table 3 jcm-12-00846-t003:** Prevention of contrast-induced nephropathy.

GFR ≥ 60 mL/min
Extremely low risk for CIN: specific prophylaxis or follow up not required
GFR < 60 mL/min (Moderate–Severe Kidney Disease)
Avoid iodinated CM whenever possible. Use iso-osmolar or low-osmolar CM at minimum possible volume. Pre- and post-hydration with isotonic saline should be considered if the expected contrast volume is > 100 mL (1 mL/kg/h 12 h before and continued for 24 h after the procedure (0.5 mL/kg/h if LVEF ≤ 35% or NYHA > 2). In statin-naive patients, pre-treatment with high-dose statins should be considered (Rosuvastatin 40/20 mg or atorvastatin 80 mg).

Abbreviations: CIN: contrast-induced nephropathy; CM: contrast medium; LVEF: left ventricular ejection fraction; NYHA: New York Heart Association. Modified from “Neumann FJ; et al. 2018 ESC/EACTS Guidelines on myocardial revascularization” [53].

**Table 4 jcm-12-00846-t004:** Premedication protocols to avoid allergic reactions.

Reaction Severity	Symptoms	Recommendation
Mild	Limited urticaria, pruritus, or skin edema; mild nasopharyngeal symptoms such as sneezing, rhinorrhea, or nasal congestion	Do not require premedication
Moderate	Generalized erythema, urticaria, pruritus, or edema Hoarseness or throat tightness with or without mild hypoxia; wheezing with mild hypoxia	Premedication is recommended Prednisone—50 mg by mouth at 13 h, 7 h, and 1 h before contrast media injection OR Methylprednisolone—32 mg by mouth 12 h and 2 h before contrast media injection PLUS Diphenhydramine—50 mg intravenously, intramuscularly, or by mouth 1 h before contrast medium
Severe	Severe edema, including facial and laryngeal edema, anaphylaxis, hypoxia	Consider alternative tests. If the test is necessary premedication is recommended Prednisone—50 mg by mouth at 13 h, 7 h, and 1 h before contrast media injection OR Methylprednisolone—32 mg by mouth 12 h and 2 h before contrast media injection PLUS Diphenhydramine—50 mg intravenously, intramuscularly, or by mouth 1 h before contrast medium

**Table 8 jcm-12-00846-t008:** Mechanical circulatory support.

	IABP	Impella (2.5, CP, 5.0)	TandemHeart	VA-ECMO
Mechanism	Diastolic augmentation of aortic pressure and improved LV performance via systolic balloon deflation (decrease in afterload)	Expels blood from LV to AO	Aspirates oxygenated blood from LA and returns to iliac artery	Drainage of deoxygenated venous blood via an extracorporeal centrifugal pump over a membrane oxygenator, and pumping back oxygenated blood to iliac artery
Indications	Consider in patients with cardiogenic shock refractory to medical therapy	Consider in patients with cardiogenic shock refractory to medical therapy	Consider in patients with cardiogenic shock refractory to medical therapy	Consider in patients with cardiogenic shock coupled with respiratory failure refractory to medical therapy
Insertion	Femoral or axillary artery to aorta	Access through femoral artery placed from LV to aorta	- Venous cannula: femoral vein to LA (requires transeptal puncture) - Arterial cannula: iliac artery	- Venous cannula: RA - Arterial cannula: iliac artery
Sheath size	7–8 Fr	13–14 Fr (2.5, CP)21 Fr (Impella 5)	15–17 Fr Arterial 21 Fr Venous	14–16 Fr Arterial 18–21 Fr Venous
Cardiac Flow	0.3–0.5 L/min	1–5 L/min	2.5–5 L/min	3–7 L/min
Duration	Weeks	7 days	14 days	Weeks
Cardiac synchrony/stable rhythm	Yes	No	No	No
Preload	---	↓↓	↓↓	↓
Afterload	↓	↓	↑	↑↑↑
MAP	↑	↑↑	↑↑	↑↑
PCWP/LVEDP	↓	↓↓	↓↓	---
Coronary perfusion	↑	↑	---	---
Complications	- Limb ischemia - Hemolysis - Thrombocytopenia - Bleeding - Infection	- Limb ischemia - Hemolysis - Bleeding - Infection	- Limb ischemia - Bleeding - Infection	- Hemolysis - Thromboembolic complications (large artificial surface) - Renal failure - Limb ischemia/amputation - Infection - Bleeding - LV overloading (may require LV decompression strategies such as septostomy, IABP, Impella, etc.) - Harlequin syndrome (upper body hypoxia from incomplete retrograde filling and oxygenation)
Contraindications	- Moderate-to-severe aortic regurgitation - Severe aortic disease	- Severe aortic stenosis - Prosthetic aortic valve - LV thrombus - VSD - Peripheral vascular disease	- Severe aortic insufficiency - Aortic dissection - Peripheral vascular disease - RV failure - VSD - Inability to tolerate systemic anticoagulation	- Severe aortic insufficiency - Aortic dissection - Inability to tolerate systemic anticoagulation

Abbreviations: AO: aorta; IABP: intra-aortic balloon pump; LA: left atrium; LV: left ventricle; LVEDP: left ventricular end diastolic pressure; MAP: mean arterial pressure; PCWP: pulmonary capillary wedge pressure; RA: right atrium; VA-ECMO: venoarterial extracorporeal membrane oxygenation; ↑: increase; ↓: decrease. Modified from “Atkinson T.M. et al.; A Practical Approach to Mechanical Circulatory Support in Patients Undergoing Percutaneous Coronary Intervention” [76].

## Data Availability

The data are available from the corresponding author on reasonable request.

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
