# Peer review of "Acute Heart Failure: Diagnostic–Therapeutic Pathways and Preventive Strategies—A Real-World Clinician’s Guide"

_jcm, 2023, doi:10.3390/jcm12030846_

Round 1

Reviewer 1 Report

In the present review the authors do a good work on putting togheter different concepts and organize current evidence regarding AHF diagnosis and treatment. I would find this more interesting if the authors manage to include a section with some perspectives derived from recent basic research derived treatment that could emerge in the next years as options for AHF management.

For example, the author include the use of Digoxin and would be nice to consider a possible substitute that could act as inotrope with lower toxicity in the future. This compound known as Istaroxime has been recently tested in a clinical trial (Carubelli 2020, Metra 2022), and there is evidence showing its low cardiotoxicity (Racioppi 2021) and capacity to reverse both sistolic and diastolic disfunction (Torre 2022)

Carubelli V, Zhang Y, Metra M, Lombardi C, Felker GM, Filippatos G, O'Connor CM, Teerlink JR, Simmons P, Segal R, Malfatto G, La Rovere MT, Li D, Han X, Yuan Z, Yao Y, Li B, Lau LF, Bianchi G, Zhang J; Istaroxime ADHF Trial Group. Treatment with 24 hour istaroxime infusion in patients hospitalised for acute heart failure: a randomised, placebo-controlled trial. Eur J Heart Fail. 2020 Sep;22(9):1684-1693. doi: 10.1002/ejhf.1743. Epub 2020 Jan 23. PMID: 31975496.

Metra M, Chioncel O, Cotter G, Davison B, Filippatos G, Mebazaa A, Novosadova M, Ponikowski P, Simmons P, Soffer J, Simonson S. Safety and efficacy of istaroxime in patients with acute heart failure-related pre-cardiogenic shock - a multicentre, randomized, double-blind, placebo-controlled, parallel group study (SEISMiC). Eur J Heart Fail. 2022 Oct;24(10):1967-1977. doi: 10.1002/ejhf.2629. Epub 2022 Aug 22. PMID: 35867804.

Racioppi MF, Burgos JI, Morell M, Gonano LA, Vila Petroff M. Cellular Mechanisms Underlying the Low Cardiotoxicity of Istaroxime. J Am Heart Assoc. 2021 Jul 20;10(14):e018833. doi: 10.1161/JAHA.120.018833. Epub 2021 Jul 3. PMID: 34219467; PMCID: PMC8483492.

Torre E, Arici M, Lodrini AM, Ferrandi M, Barassi P, Hsu SC, Chang GJ, Boz E, Sala E, Vagni S, Altomare C, Mostacciuolo G, Bussadori C, Ferrari P, Bianchi G, Rocchetti M. SERCA2a stimulation by istaroxime improves intracellular Ca2+ handling and diastolic dysfunction in a model of diabetic cardiomyopathy. Cardiovasc Res. 2022 Mar 16;118(4):1020-1032. doi: 10.1093/cvr/cvab123. PMID: 33792692; PMCID: PMC8930067.

Reviewer 2 Report

For a review in an internal medicine journal, this review of acute heart failure is not overly specialized, but it covers all the important aspects, including actual testing procedures, drug dosing, and a brief description of the mechanism of action of the drugs. In addition to drugs, various other devices are described, including their mechanisms of action.

However, there are a few areas that seem to require some additional description.

 There is a mention in line 339 regarding B-lines as an evaluation of pulmonary congestion. I think it would be better to have a more detailed description regarding that evaluation method.

 Although there is some overlap with the criteria for endotracheal intubation, I would suggest adding an absolute contraindication for NIV in the section on mechanical ventilation in 4.2.1.

 In the area of prevention strategies, why not mention alcohol consumption, treatment of anemia, and infection control measures, including vaccination?
